# Influence of Shielding Gas on Microstructure and Properties of GMAW DSS2205 Welded Joints

**DOI:** 10.3390/ma14102671

**Published:** 2021-05-20

**Authors:** Xin-Yu Zhang, Xiao-Qin Zha, Ling-Qing Gao, Peng-Hui Hei, Yong-Feng Ren

**Affiliations:** 1Luoyang Ship Material Research Institute, Luoyang 471023, China; 18629660516@163.com (X.-Y.Z.); Gaolq@163.com (L.-Q.G.); 2Henan Key Laboratory of Technology and Application of Structural Materials for Ships and Marine Equipments, Luoyang 471023, China; 3Luoyang Sunrui Special Equipment Co., Ltd., Luoyang 471000, China; heipenghui@163.com; 4Linzhou, Fengbao Pipe Industry Co., Ltd., Anyang 456592, China; 18262075136@163.com

**Keywords:** 2205 duplex stainless steel (DSS 2205), welded joint, shielding gas, pitting corrosion, intergranular corrosion

## Abstract

In the present study, the microstructures and properties of DSS 2205 solid wire MIG welded samples prepared in different shielding gases (pure Ar gas, 98%Ar + 2%O_2_ and 98%Ar + 2%N_2_) were investigated for improving the weldability of DSS 2205 welded joint. The work was conducted by mechanical property tests (hardness and tensile test) and corrosion resistance property tests (immersion and electrochemical tests). The results show that adding 2%O_2_ into pure Ar gas as the shielding gas decreases crystal defects (faults) and improves the mechanical properties and corrosion resistance of the welded joints. Phase equilibrium and microstructural homogeneity in welded seam (WS) and heat-affected zone (HAZ) can be adjusted and the strength and corrosion resistance of welded joints increased obviously by adding 2%N_2_ to pure Ar gas as the shielding gas. Compared with DSS 2205 solid wire MIG welding in 98%Ar + 2%O_2_ mixed atmosphere, the strength and corrosion resistance of welded joints are improved more obviously in 98%Ar + 2%N_2_ mixed atmosphere.

## 1. Introduction

Duplex stainless steel (DSS) composed of both austenite (γ) and ferrite (α) has a combination of excellent corrosion resistance and mechanical strength, which is used for petroleum, nuclear and chemical industries [1]. Duplex Stainless Steels 2205 (DSS 2205), as the third generation of duplex stainless steel [2], compared with the latest DSS (LDX 2404 lean duplex stainless steel), has superior local corrosion resistance [3], and is widely used in the structural parts of coastal building, bridge and offshore platform because the excellent pitting corrosion and intergranular corrosion resistance [4,5,6]. For the complex structural parts of duplex stainless steel, welding is probably the most common joining method. It’s known that the quality of the welded joint plays the most critical role in the service reliability of welded structural parts. However, there are great differences of the microstructure and mechanical properties between welded joint and base metal (BM). Therefore, it has great significance to investigate the microstructure and corrosion properties of the welded joint in DSS 2205.

At present, the welding process suitable for DSS 2205 mainly includes manual metal arc welding, shielded metal arc welding, metal inert gas (MIG) shielded welding and so on. Among the above, MIG welding has the advantage of simple operation, low cost and broad application [7]. There are two categories of MIG welding, named solid wire MIG welding and fluxcored wire MIG welding, respectively. Compared with flux-cored wire MIG welding, solid wire MIG welding has many benefits, such as lower cost, fewer welding defects, but the weldability is inferior [8]. Therefore, how to improve the technology of solid wire MIG welding has received significant attention. Suban [9] et al. improved the processability of solid wire MIG welding joints by using different shielding gases. The results showed that the mixture of argon and carbon dioxide had a good effect on improving the production efficiency and welded joint performance of solid wire MIG welding. García et al. [10] investigated the distinction of electrochemical parameters of DSS 2205 welded plates between two different shielding gases (98%Ar + 2%O_2_ and 97%Ar + 3%N_2_). The results showed 50/50 in the α/γ phase ratio in welded joint could be obtained with the simultaneous application of a 3 mT EMF during the welding process, and the growth of detrimental phases would be hindered. In addition, they proved the application of 3 mT during GMAW of DSS enhanced the resistance to localized corrosion of the welded joint in the welded zone when compared with welds made using 97%Ar + 3%N_2_ shielding gas mixture. To research how hydrogen affects the properties of DSS, Świerczyńska et al. [11] studied the corrosion behavior of hydrogen charged super duplex stainless steel welded joints. The results showed that base metal had the best corrosion resistance, and with the increase in hydrogen content, the pitting and general corrosion resistance of HAZ and welded zone decreased. On this basis, they also researched the hydrogen embrittlement of super duplex stainless steel welded joint under cathodic protection [12]. The results showed that the hydrogen embrittlement sensitivity of the welded joint of super duplex stainless steel significantly increased, and brittle fracture appeared under the condition of cathodic protection in artificial seawater. Gurcik et al. [13] discussed the impacts of different shielding gases (Ar, CO_2_, O_2_ and He) on the shape of the welded joint and welding productivity. The results showed that increasing the content of O_2_ or decreasing the contents of CO_2_ and He can adjust the shape of welded joint, reduce the residual stress of the welded structure, and increase welding production efficiency. Numerous studies have shown that choosing mixed gases as the shielding gas has substantial advantages to improve the solid wire MIG welding properties and performance. A few articles were published related to mixed gases of Ar-CO_2_ [14,15,16,17]. Despite these progresses, welding under gas mixtures of Ar-O_2_ and Ar-N_2_ has been not well explored regarding property enhancements, especially corrosion resistance. At present, the research on the corrosion resistance of duplex stainless steel is mainly to analyze the influence of microstructure changes on the corrosion resistance of duplex stainless steel. Jerzy et al. [18] studied the influence of microstructure changes on the corrosion resistance of DSS2205 with aging treatment at 500 °C and 700 °C. At 500 °C, the hardness of the sample increased with aging time, which may be due to the formation of α phase, and the stress corrosion sensitivity of DSS2205 increased. At 700 °C, as the aging time increases, DSS2205 secondary austenite γ phase was existed, and the stress corrosion sensitivity of DSS2205 did not change significantly. In addition, the current research on duplex stainless steel welded joint is mainly to analyze the influence of welding process on its corrosion resistance by changing crafts. Marek et al. [19] studied the effects of heat input and nitrogen content in the welding process on corrosion resistance of duplex stainless steel, and the results showed that the appropriate increase in heat input energy would improve the microstructure uniformity and corrosion resistance of duplex steel welded joints, and the addition of 2–4% N2 would also improve the corrosion resistance of the welded joints. It can be seen that the research on the corrosion resistance of DSS2205 welded joints has been relatively systematic, but there are still some aspects that need to be supplemented.

In this work, the microstructures and the properties (especially corrosion resistance) of the DSS 2205 welded joints prepared in three different atmospheres were studied, which give the technology support for shielding gases standard and the application for solid wire MIG welding.

## 2. Materials and Methods

The base metal used in the current study was commercial DSS 2205 plates with a thickness of 6 mm (500 × 250 × 6 mm), and the MIG welding of DSS 2205 was performed by using ER2209 solid wire with a diameter of 1.2 mm. The specific chemical composition of DSS 2205 plate and ER2209 solid wire are detailed in Table 1 [20].

The experiments in this study were divided into three groups. The first group was welded with pure argon as the shielding gas (No.1); the other two groups were welded with shielding gas mixtures: 98%Ar + 2%O_2_ (No.2) and 98%Ar + 2%N_2_ (No.3), respectively. Before welding, a “V” groove with an angle of 60° was made on the joint of two steel plates. Automated MIG welding of 2205 duplex stainless steel plate was carried out by three-layer welding process. A backing bar was used during the MIG welding process. The welding current was 160~220 A, arc voltage 27 V, welding speed 100~160 mm/min and gas flow rate 12~15 L/min, and contact tip to work distance (CTWD) 15 mm.

In order to evaluate the susceptibility of the different microstructural components to pitting, pitting corrosion immersion testing was carried out according to ASTM A923 C-2006 of the U.S. The corrosion testing was carried out by immersing samples with the dimensions of 30 mm × 20 mm × 3 mm in a 10 wt% FeCl_3_ solution at a test temperature 25 ± 1 °C for 24 h. The property of corrosion resistance was characterized by corrosion rate which can be calculated by the following corrosion rate Equation:(1)$$K=\frac{240\times (g_{0}-g_{1})}{S\times t}$$
where *K* is the corrosion rate [mg/(dm^2^·day),mdd], *g*_0_ is the specimen mass before corrosion (g), *g*_1_ is the specimen mass after corrosion (g), *S* is the specimen superficial area (m^2^), and *t* is corrosion time (h).

The Gill AC Bi-STAT electrochemical workstation was used to evaluate the electrochemical corrosion performance of DSS 2205 solid wire MIG welded samples prepared in different shielding gases. Pitting corrosion electrochemical testing was carried out according to GB/T 24196-2009 of China. The samples with a dimension of 10 mm × 10 mm were selected to be near-surface on welded seam (WS) of joint. The test was conducted in 3.5% NaCl solution with 40(±1) °C. Before the test, the open circuit potential (E_ocp_) was recorded by exposing the specimen in the solution for about 60 min. When potential was stable, the test was started at the potential of −150 mv/SCE+E_ocp_ with a scanning rate of 0.833 mV/s. The scanning range was from −150 mv/SCE+E_ocp_ to 200 mV/SCE+E_p_ (pitting corrosion potential). The results were disposed by OriginLab after test, then the polarization curves were obtained to calculate electrochemical parameters.

According to ASTM A262-15 of the U.S., intergranular corrosion immersion testing was carried out by immersing samples with the dimensions of 30 mm × 20 mm × 3 mm in the 3 wt%(Fe)_2_(SO_4_)_3_-50 wt%H_2_SO_4_ solution boiling about 120 h. The property of corrosion resistance was characterized by corrosion rate and photographs at the polished sections. The corrosion rate was calculated by Equation (1). The double loop electrochemical potentiokinetic reactivation (DL-EPR) method was performed to evaluate the samples’ susceptibility to intergranular corrosion. The tests were carried out according to ASTM G108-1994 (2005), the Gill AC Bi-STAT electrochemical workstation was conducted to test DL-EPR specimens in each group. The samples were taken from near surface in welded joints of 10 × 10 mm. The electrolyte consisted of 2 mol H_2_SO_4_ + 1 mol HCl at 30(±1) °C. Cyclic potentiokinetic polarization was conducted from the open circuit potential to 200 mV vs. SCE with a scanning rate of 1 mV/s. The scanning range was from 200 mV/SCE+E_ocp_ to 200 mV/SCE+E_p_. The results were disposed by OriginLab after test, then DL-EPR curves were obtained. The samples were electrolytically etched to observe the metallographic structure of welded joint after DL-EPR test. Hardness was measured using HVS10B Vickers indenter with 10 kg load and 10 s holding time. Locations of test points are shown in Figure 1 [20]. Tensile test specimens in accordance with GB/T 2651-2008, with a gauge length of 75 mm, gauge width of 18 mm, and thickness of 6 mm, were prepared as shown in Figure 2 [20]. Since the properties of the welded joint can be considered to be symmetrically distributed to the sides of the weld axis, thus, only one side of the weld axis was selected to test. The tensile tests were carried out on the tensile workstation (Instron5587), and the loading speed was set to 0.5 mm/min.

The metallographic microstructure in welded seam (WS) and heat-affected zone (HAZ) was observed by scanning electron microscope (SEM). The metallographic specimens’ preparation was carried out following standard procedures for microstructure characterization. Sodium sulfite hydrochloric acid solution was used as an etchant to reveal the ferrite and austenite phases. The ferrite content was measured by metallographic image analysis software, ten photos (100×) at least in each group. The transmission samples were cut into slices of 0.5 mm along the weld axis, and then prepared by the method of double jet electrolysis and ion-beam thinning. Finally, specimen microstructure was amplified to analyze in the TEM (CM200) with 200 kV accelerated voltage.

## 3. Results

### 3.1. Microstructure

Figure 3 shows the metallographic microstructure of WS and HAZ in DSS 2205 solid wire MIG welding joints prepared in different shielding gases. As shown in Figure 3, it is clear that the microstructure characteristics of the welded joints in three groups are same, that are all composed of ferrite (black region) and austenite (white region). However, for one welded joint, the microstructure characteristics of WS and HAZ are different. In the WS, intergranular austenite, intracrystalline austenite and Widmanstätten austeniteare primary depositions, and the white regionsare typical cast structure. However, in the HAZ, only intergranular austenite and intracrystalline austenite are formed, and the austenite content is lower than that of WS (Figure 3). Comparing ferrite content of samples in three groups (refer Table 2), it is noticed that ferrite contents of No.1 and No.2 are almost the same. The ferrite contents of No.1 and No.2 samples are 58.54% and 57.18% in the WS, and are 65.17% and 64.58% in the HAZ, respectively. The ferrite content of No.3 in WS and HAZ, respectively, are 51.35% and 59.24%, both are lower than that of other two groups.

Figure 4 shows the TEM images of DSS 2205 solid wire MIG welding joints and the diffraction patterns of ferrite phase (α phase) and austenite (γ phase). It can be observed that no phase is precipitated in α and γ phase, even interface between α and γ phase. However, there are some dislocations in α and γ phase obviously, especially for the samples of No.2 and No.3 (Figure 4b,c). In contrast, a large number of stacking faults occur in the interface between α and γ phase closing to α phase in the sample of No.1. In addition, the dislocation density of α phase is higher than that of γ phase. Due to more slip systems in ferrite than austenite, dislocation tangling in ferrite are easier created than in austenite under the same deformation conditions. This behavior results in different deformation in the biphase interface of α and γ phase. After that, dislocation pile-up occurs in this region inevitably.

### 3.2. Mechanical Properties

Figure 5 shows the hardness distribution of DSS 2205 solid wire MIG welding joints prepared in different shielding gases. As shown in Figure 5, the hardness of No.3 sample remains stable, changing little from BM to WS, while the hardness of No.1 and No.2 samples both gradually decreases from BM to WS, and the lowest hardness appears in the weld seam. In addition, the hardness of No.2 sample, by contrast, is higher than that of No.1 sample in both HAZ and WS.

Figure 6 shows fracture positions of tensile specimens of DSS 2205 solid wire MIG welding joints under different shielding gases. As can be seen in Figure 6, the fracture positions of tensile specimens in three groups are both in WS. In general, the lower the hardness, the lower the strength. This is a good explanation for the fracture of No.1 and No.2 samples appeared at the WS, because of the lowest hardness of WS. However, for No.3 sample, the reason why the fracture appears at the WS may be that the BM zone is close to clamps which would reinforce this region; thus, the fracture position is also in WS, although the hardness of WS is equivalent to that of BM. The tensile strength values of No.1, No.2 and No.3 samples are 803 MPa, 812 MPa and 823 MPa, respectively. That is consistent with the results of hardness. Therefore, adding 2%O_2_ or 2%N_2_ into pure Ar gas as the shielding gas can improve hardness and tensile strength of DSS 2205 solid wire MIG welding joints. Among above, adding 2%N_2_ performed has the best effect.

### 3.3. Corrosion Resistance

#### 3.3.1. Pitting Corrosion

The corrosion rate of DSS 2205 solid wire MIG welding joints under different shielding gases were obtained by chemical immersion test. Figure 7 shows the results of the corrosion rate using a bar chart. In Figure 6, the corrosion ratesof No.1, No.2 and No.3 samples are 3.48 mdd, 3.29 mdd and 1.36 mdd, respectively. The results indicate that adding 2%O_2_ or 2%N_2_ into pure Ar gas can leads to the reduction of corrosion rate. Additionally, the corrosion rate reduces significantly when adding 2%N_2_ into pure Ar gas as the shielding gas.

Figure 8 shows polarization curves of DSS 2205 solid wire MIG welding joints prepared in different shielding gases, and the experimental results of electrochemistry are listed in Table 3. From Figure 8 and Table 4, the corrosion potential (E_Corr_) of No.2 sample is the highest of −255 mV, and the E_Corr_ of No.1 sample (−286 mV) is close to that of No.3 sample (−291 mV). These results suggest that adding 2%O_2_ into the pure Ar gas can lead to an increase in the E_Corr_. This is because the formation of compact metal oxide film on the WS surface with O_2_ addition reduces the initial corrosion activity of sample surface. The pitting potential (E_p_) of No.1, No.2 and No.3 samples are 1025 mV, 1050 mV and 1103 mV, respectively. Compare with No.1 sample, the E_p_ of No.2 and No.3 samples both increases due to the addition of O_2_ or N_2_ into pure Ar gas as mixed shielding gas. E_p_ is the potential at which the metal oxide film begins to be damaged with increasing of scanning potential. Based on this, the larger the E_p_, the better the pitting corrosion resistance. Considering the pitting corrosion resistance of these three samples, the No.3 sample is the best, followed by No.2 and No.1 samples. This consequence is consistent with of immersion test.

Furthermore, E_Corr_ and E_p_ are both related to property of pitting corrosion resistance, so the potential difference (ΔE = E_p_ − E_Corr_) is conducted on characterizing pitting corrosion resistance property scientifically [21]. Obviously, a larger ΔE indicates better pitting corrosion resistance. Table 3 presents the potential difference of the samples in three groups. From Table 4, the ΔE value of No.1, No.2 and No.3 samples are 1311 mV, 1305 mV and 1394 mV, respectively. The ΔE of No.3 sample is significantly higher thanthat of No.1 and No.2 samples, which can confirm further that adding 2%N_2_ into pure Ar gas as the shielding gas improves the pitting corrosion resistance significantly.

#### 3.3.2. Intergranular Corrosion

The intergranular corrosion of DSS 2205 solid wire MIG welding joints under different shielding gases were obtained by chemical immersion test. Figure 9 shows the results of the corrosion rate using a bar chart. It can be found that the corrosion rates of No.1, No.2 and No.3 samples are 337.52 mdd, 329.59 mdd and 314.42 mdd, respectively, which is confirmed that the corrosion rate of welded joint is reduced by adding 2%O_2_ or 2%N_2_ into pure Ar gas. By contrast, the effect of adding 2%N_2_ on the intergranular corrosion rate is significantly higher than that of adding 2%O_2_.

Figure 10 shows the specimens’ cross section morphology of the samples after intergranular corrosion test. There is no intergranular corrosion feature on the sample surface, but there are general heterogeneous corrosion pits because pits morphology can be seen in some areas. It is confirmed that DSS 2205 welded joints are not sensitive to intergranular corrosion according to the standard test results. This indicates that the intergranular corrosion rates in column graph (Figure 9) are general heterogeneous corrosion rates. The double-loop electrochemical potentiokinetic reactivation (DL-EPR) techniques were used to evaluate the susceptibility to intergranular corrosion of DSS 2205 solid wire MIG welding joints under different shielding gases. The DL-EPR curves in the WS zone of MIG welding joints can be seen in Figure 11. It is clearly observed that anode activation peak current densities of No.1, No.2 and No.3 samples are 3.74 mA/cm^2^, 3.16 mA/cm^2^ and 2.23 mA/cm^2^, respectively. Additionally, reactivation peak current densities of the three samples are roughly the same, about 0.24 mA/cm^2^. To further determine whether intergranular corrosion occurred after DL-EPR test, microstructure observation was conducted on the surface of samples in each group (Figure 12). On the surface of No.1 sample, there are some general heterogeneous corrosion obviously, but there is no intergranular corrosion feature. However, there are few signs of corrosion occurred on the surface of No.2 and No.3 samples, only ferrite and austenite microstructuresareobserved obviously. The results are in good agreement with those of the chemical immersion method.

## 4. Discussion

During the process of DSS 2205 solid wire MIG welding, the microstructure of WS firstly transforms into high-temperature ferrite; then, the austenite phase nucleated and grew in grain boundary and intragranular ferrite, and finally the intergranular austenite and the intracrystalline austenite were formed in the cooling process. In addition, Widmanstätten austenite are formed because the cooling rate is too fast. The microstructure of HAZ is formed by the rapid heating and cooling process of the BM. Due to the fast heating and cooling rate, the ferrite-to-austenite transformation was not sufficient. Therefore, the austenite mainly nucleated along the ferrite grain boundary rather than in ferrite grains, and a large amount of ferrite lamellae is remained [22]. Because O_2_ is an excellent oxidizing agent in shielding gas that increases the temperature of molten pool, adding 2%O_2_ into pure Ar gas increases the metal fusibility of WS, thus improving the appearance of the weld and reducing the welding defects [23]. However, the excess oxygen will show strong oxidizer, which increases heat production and uneven microstructure in WS, and even produces more defects. Nakamura et al. [16,24] studied the influence of O_2_ content in shielding gas on the properties of welded joints, and found that 98%Ar + 2%O_2_ used as the shielding gas had the best effect on improving the uniformity of weld microstructure and improving the service performance of welded joints. It is consistent with the conclusion of this work. In this study, the faults in transmission microstructure of No.2 sample are disappeared. However, the ferrite content of No.1 sample is almost the same as that of No.2 sample. It indicates that the addition of 2%O_2_ into pure Ar gas can not effectively reduce the cooling rate of WS and HAZ, and does not promote the ferrite-to-austenite transformation. Nitrogen is an indispensable element of austenite, and beneficial for the ferrite-to-austenite transformation [25,26]. Varbai et al. [27,28] studied the influence of weld thermal cycles and shielding gas nitrogen content on DSS weld microstructure by simulated experiment. The results showed that the nitrogen loss from the molten pool caused a lower austenite fraction in the weld metal, and higher nitrogen content in the shielding gas will result in higher initial austenite fraction. Therefore, adding 2%N_2_ into pure argon as the shielding gas can replenish the N element lost in welding process and promote the ferrite-to-austenite transformation. Thus, for the No.3 sample, the ferrite content in WS and HAZ is effectively reduced, the phase equilibrium of the microstructure is improved, and the crystal defects are inhibited. Besides, the dislocation density of No.3 sample is higher than that of No.2. After analysis, the nitrogen atoms can create Cottrell atmosphere and cause the pinning effect on dislocation line, which block dislocation movement and cause dislocation pile-up [29,30].

The hardness of No.1 and No.2 samples both reduced from BM to WS. On the one hand, it is because the carbon content of welding wire is lower than that of BM (Table 1). On the other hand, it is also related to the incomplete formation of austenite and the faults in WS and HAZ. Controlling the appropriate ferrite-austenite ratio (1:1) is an excellent method to improve the strength and toughness of dual-phase stainless steel [31]. The crystal defects can reduce the strength of welded joints because it promotes dislocation movement and crack initiation in tensile process [32,33]. It follows from the above that the hardness of No.2 sample is higher than that of No.1 due to the disappearance of faults in microstructure, and the hardness of No.3 sample is almost unchanged, which benefits by adding 2%N_2_ into pure Ar gas that can promote the ferrite-to-austenite transformation and obtain the appropriate ferrite-austenite ratio in WS and HAZ (Table 2). Furthermore, the disappearance of fault in microstructure and increasing of dislocation density can also improve sample’s hardness due to the dislocation pile-up [34], which can cause the strengthening of materials, thus, No.3 welded joint has the highest strength.

The formation of crystal defects (faults) not only reduces the strength of welded joins, but also becomes the initial points of corrosion, whichpromotes the occurrence of pitting corrosion [35,36,37]. This is the reason why the corrosion resistance of No.1 sample is the worst in both the pitting corrosion test and the intergranular corrosion test. After adding 2%O_2_ into pure Ar gas, the stacking faults disappear and the metal oxide film becomes denser (higher E_Corr_), and the corrosion resistance is improved. In addition, after adding 2%N_2_ into pure Ar gas, the ferrite-to-austenite transformation is promoted, the phase equilibrium and microstructure homogeneity of WS and HAZ are improved, and the corrosion resistance of welded joints is significantly increased.

## 5. Conclusions

In this paper, the microstructure, mechanical property and corrosion resistance of 2205 duplex stainless steel MIG welded joints under three shielding gases were investigated. The main conclusions are listed as follows.

When DSS 2205 solid core MIG welding is performed in 98%Ar + 2%O_2_ mixed atmosphere, the crystal defects (layering faults) are reduced, and the joint strength and corrosion resistance are improved, although the ferrite content of welded joints is similar to that of welded joints in pure Ar atmosphere.When DSS 2205 solid core MIG welding is performed in 98%Ar + 2%N_2_ mixed atmosphere, the austenite contents in WS and HAZ are both increased. The phase equilibrium and microstructure homogeneity are improved, and the strength and the corrosion resistance of the welded jointsare also enhanced.Compared with DSS 2205 solid core MIG welding in 98%Ar + 2%O_2_ mixed atmosphere, the strength and corrosion resistance of welded joints are improved more obviously in 98%Ar + 2%N_2_ mixed atmosphere.

## Figures and Tables

**Figure 1 materials-14-02671-f001:**
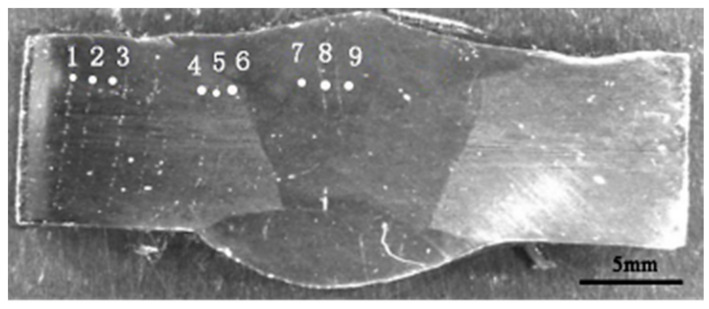
Hardness point location.

**Figure 2 materials-14-02671-f002:**
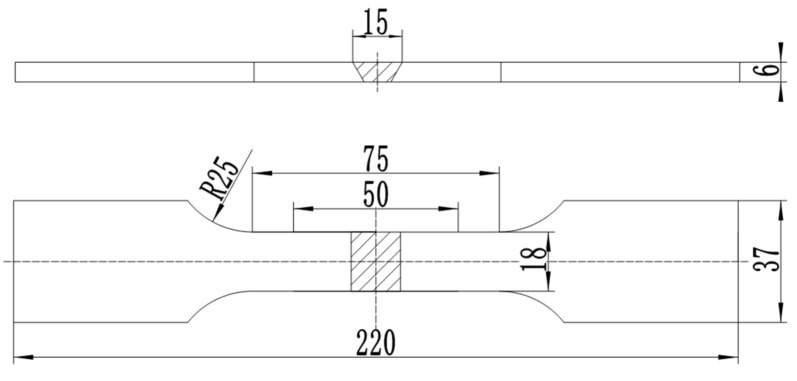
Schematic diagram of plate tensile specimen.

**Figure 3 materials-14-02671-f003:**
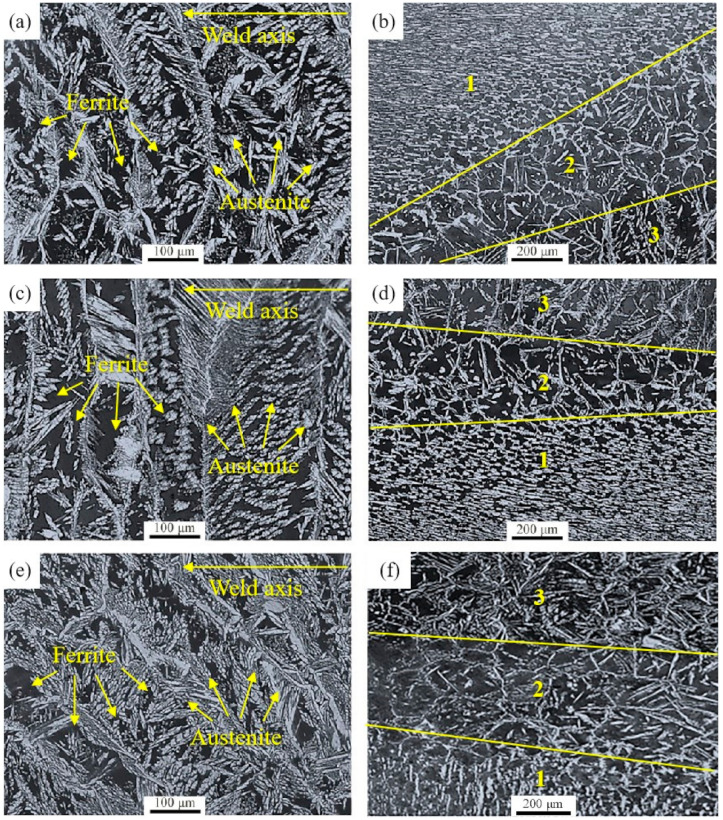
Metallographic structure of DSS 2205 solid wire MIG welding joints prepared in different shielding gases. (**a**,**b**) WS and welding joints region (pure Ar gas); (**c**,**d**) WS and welding joints region (98%Ar + 2%O_2_); (**e**,**f**) WS and welding joints region (98%Ar + 2%N_2_); (1) BM (2) HAZ (3) WS.

**Figure 4 materials-14-02671-f004:**
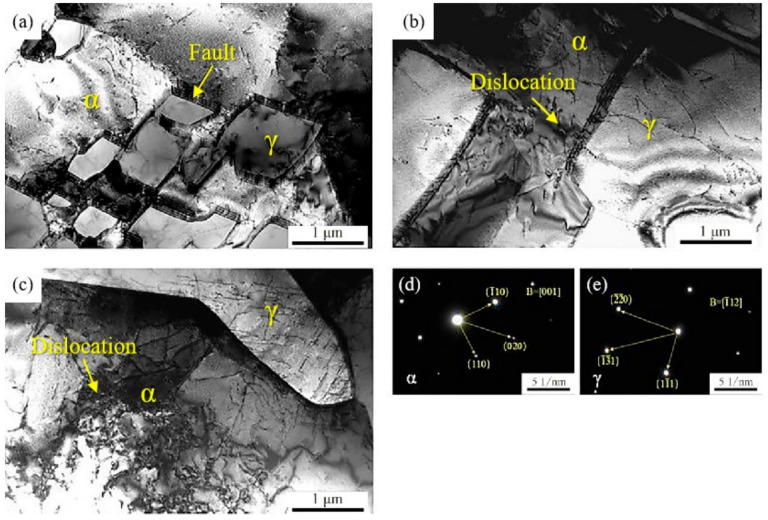
TEM images of DSS 2205 solid wire MIG welding joints prepared in different shielding gases. (**a**) Ar; (**b**) 98%Ar + 2%O_2_; (**c**) 98%Ar + 2%N_2_; (**d**) ferrite diffraction pattern; (**e**) austenite diffraction pattern.

**Figure 5 materials-14-02671-f005:**
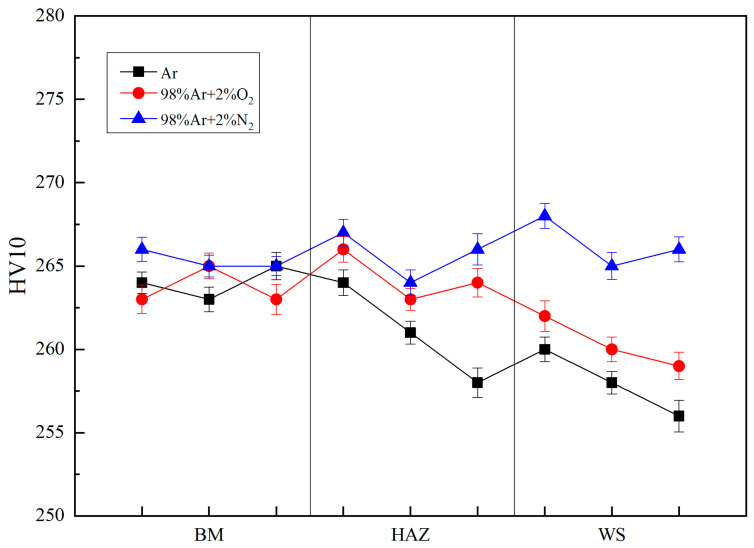
Hardness distribution chart.

**Figure 6 materials-14-02671-f006:**
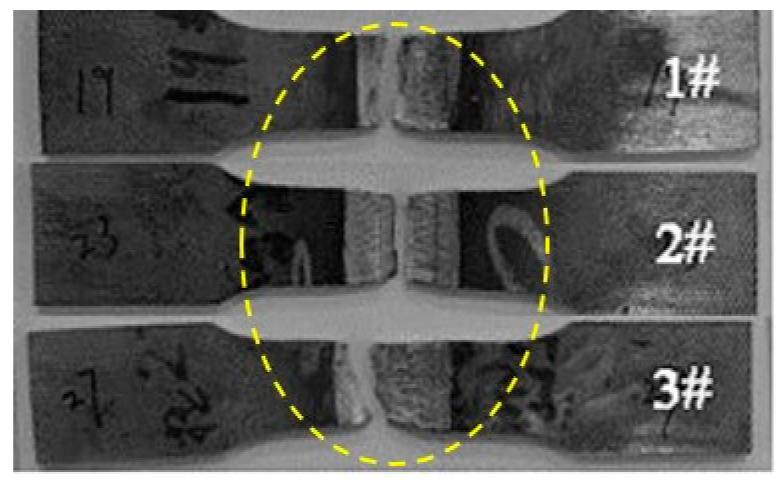
Fracture position of flat tensile specimens of DSS 2205 solid wire MIG welding joints prepared in different shielding gases. 1#-Ar; 2#-98%Ar + 2%O_2_; 3#-98%Ar + 2%N_2_.

**Figure 7 materials-14-02671-f007:**
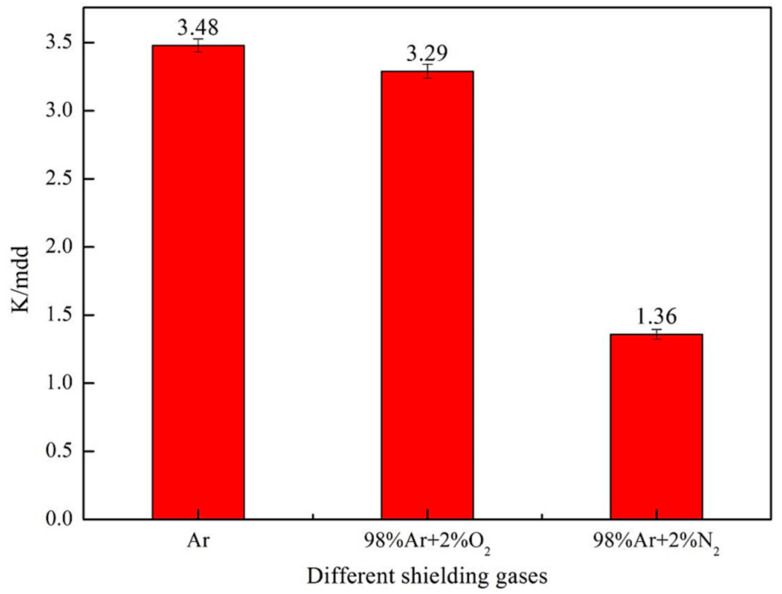
Pitting corrosion rate of welded joint under different shielding gases.

**Figure 8 materials-14-02671-f008:**
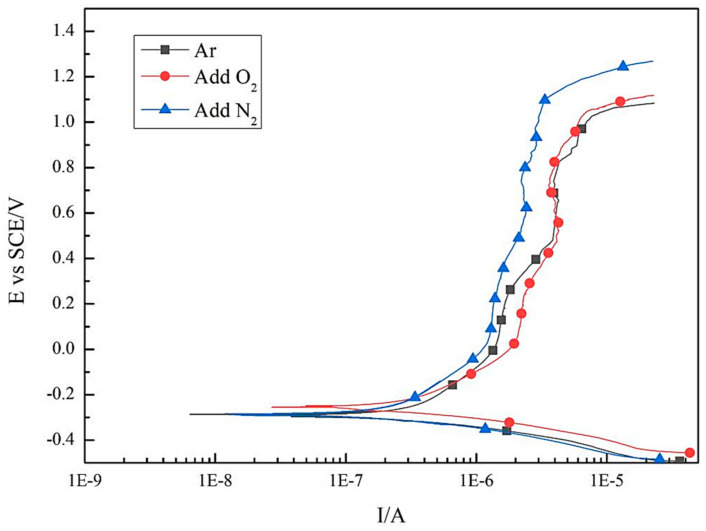
Polarization curves of DSS 2205 solid wire MIG welding joints under three different shielding gases.

**Figure 9 materials-14-02671-f009:**
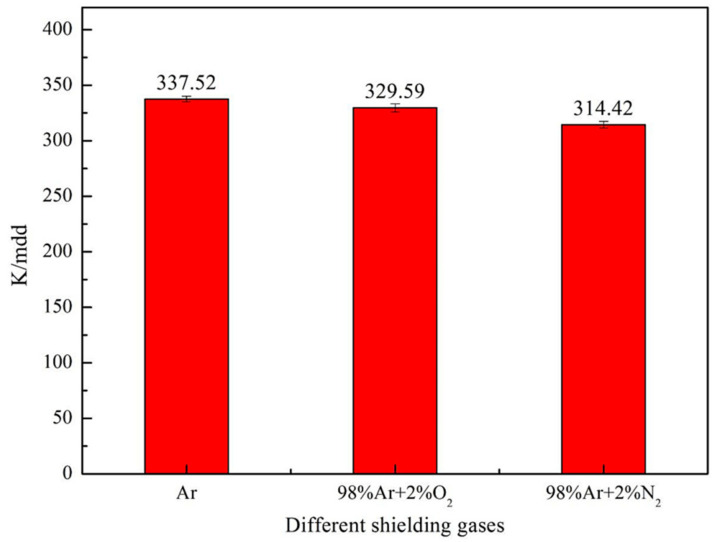
Intergranular corrosion rate of the welded joints under different shielding gases.

**Figure 10 materials-14-02671-f010:**
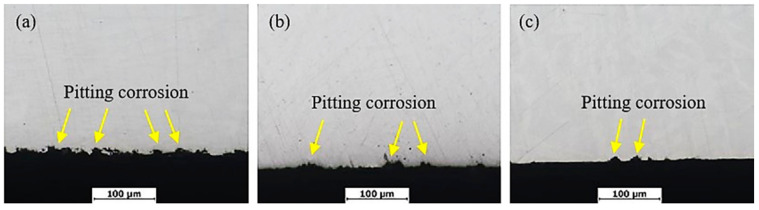
Cross section morphology of intergranular corrosion for the welded joints under different shielding gases. (**a**) 1#-Ar; (**b**) 2#-98%Ar + 2%O_2_; (**c**) 3#-98%Ar + 2%N_2_.

**Figure 11 materials-14-02671-f011:**
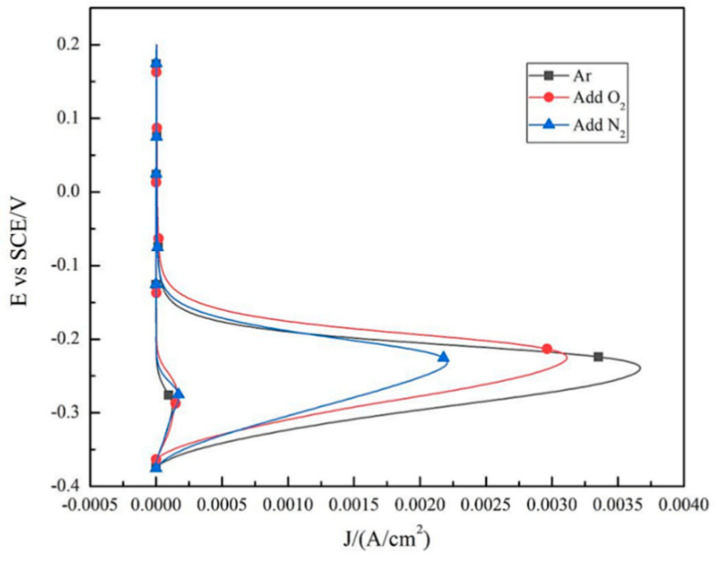
DL-EPR test for the welded joints under different shielding gases.

**Figure 12 materials-14-02671-f012:**
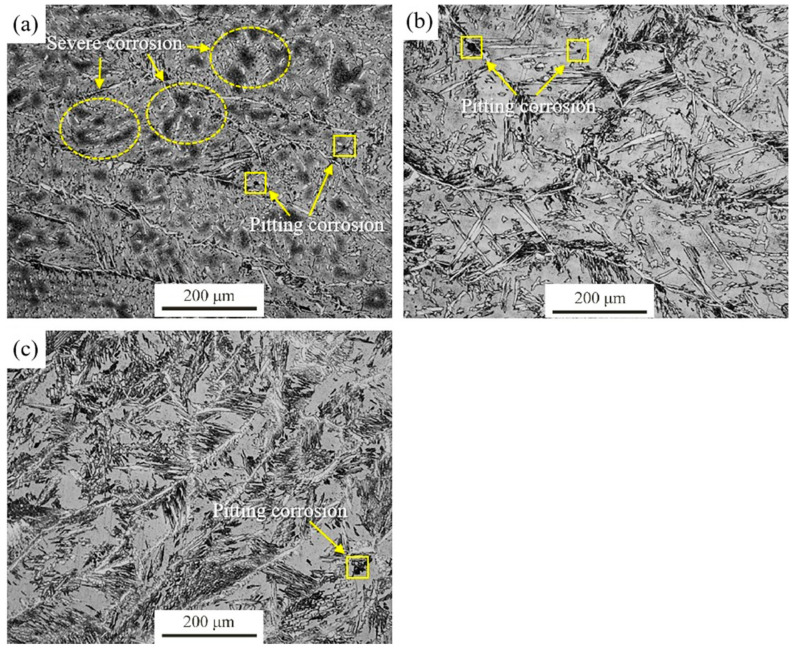
Metallographic structure of the welded joints under different shielding gases after DL-EPR test (**a**) 1#-Ar; (**b**) 2#-98%Ar + 2%O_2_; (**c**) 3#-98%Ar + 2%N_2_.

**Table 1 materials-14-02671-t001:** Chemical composition of DSS 2205 plate and ER2209 solid wire (wt%).

Material	C	Si	Mn	Cr	Ni	Mo	N	Fe
DSS2205	0.025	0.60	1.50	22.50	5.70	3.00	0.15	Balance
ER2209	0.017	0.57	1.61	22.06	8.84	2.68	0.11	Balance

**Table 2 materials-14-02671-t002:** Ferrite content of WS and HAZ in different shielding gases (wt%).

Samples	No.1	No.2	No.3
Ferrite content in WS (%)	58.54	57.18	51.35
Ferrite content in HAZ (%)	65.17	64.58	59.24

**Table 3 materials-14-02671-t003:** The tensile strength values of No.1, No.2 and No.3 samples.

Samples	No.1	No.2	No.3
Tensile strength (MPa)	803	812	823

**Table 4 materials-14-02671-t004:** Potential difference of welded joint under three different shielding gases.

Samples	Corrosion Potential E_Corr_ (mV)	Pitting Potential E_p_ (mV)	Potential Difference ΔE (mV)
No.1	−286	1025	1311
No.2	−255	1050	1305
No.3	−291	1103	1394

## Data Availability

Data available in a publicly accessible repository.

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
