# Peer review of "Influence of Shielding Gas on Microstructure and Properties of GMAW DSS2205 Welded Joints"

_materials, 2021, doi:10.3390/ma14102671_

Round 1

Reviewer 1 Report

  • It is mentioned in the introduction that DSS2205 steel is the third generation of this class. However, its advantage over previous generations is not mentioned. Modern duplex steels of the LDX type with reduced Ni content are not mentioned.
  • Giant. 3 (b), (d), (f) - I recommend adjusting the identical orientation of the marked weld zones
  • Figure 4 (d), (e) Instead of a diffraction pattern, it would be better to give an EBSD analysis where the phases in the microstructure are clear
  • Figure 5 A graph of the macro hardness dependence (a) and a small image of the broken tensile bodies (b) are combined. I recommend showing the result of the tensile test in the next graph and with a photograph of the result of the tensile test.
  • The title of the article refers to the properties of DSS2205, but only corrosion properties are discussed in the conclusion (strength is not mentioned). I recommend editing either the title or the conclusions.

Reviewer 2 Report

Dear Authors,   I have read you manuscript "Impact of shielding gases on microstructures and properties of DSS2205 joints under solid wire metal insert gas welding" with interest. I have some suggestions, which may be helpfull in improving your work.  

  1. Line 26  - please extend this part of introduction to avoid citation of 4 work in one bracket [2-5].
  2. Line 37 - citation of article No. 7 is not correct, it concerns a completely different welding method
  3. Lines 48 and 50 you have used the term welded seam but in that context I suggest to use just welded joint.
  4. Line 56 I suggest to comment even short the influence of other important factors on corrosion resistance welded joints of DSS 2205 as below:
    https://doi.org/10.26628/ps.v90i5.916
    https://doi.org/10.2478/pomr-2014-0047
  5. Line 75 - specified welding parameters have got to big range "160-220 A" (by the way voltage have to be in range too consequently) "100-160 mm/min". There is no doubt that is significant influence of heat input of welding process on corrosion resistance of DSS 2205. Due to the elimination of the influence of the energy parameters of the welding process and considering only the type of shielding gas, it is necessary to precisely describe and confirm the welding parameters and heat input. Otherwise there is no point in considering the effects of shielding. DSS 2205 steel is very specifically sensitive on heat cycle which should be precisely described.
  6. Line 166. There is no "macrohardness", a jargon unacceptable in scientific article. 
  7. On Fig. 1 you have presented single hardness imprint. Thus, how was the mean value calculated, and even more so, the standard deviation marked in Fig. 5. How do you explain the low standard deviation of hardness in HAZ (after recrystallization) and in the weld (primary crystallization). Without a detailed description of the hardness distribution method, it seems statistically unreliable. 
  8. Fig 5b - the picture is blurred and unreadable
  9. Line 259. "solid core MIG" term is incorrect. Core MIG wire is with powder inside. I suggest to leave solid wire MIG only.

General remark. 
It seems that the authors did not appreciate the fact that when welding dss2205 steel, inappropriate welding energy parameters may be more important than a small addition of oxygen or carbon dioxide.
Welded joints od DSS2205 should be made in the same energy conditions and in the correct heat input range.

Regards

Reviewer 3 Report

The paper seem interesting but the number of tested samples is not indicated in the text and seems to be only one per test. If that's the case, the results are too close for the different groups different to draw any definitive conclussions.

For example, tensile test give 803, 812 and 823 MPa for the 3 different groups. This small difference could be easily attributed to the statististical dispersion inherent to the tensile test.

So, a word of caution should be provided in the text regarding the validity of results if only one sample pre group was tested. More tests could confirm the conclussions presented in the paper.  If the presented values correspond to a series of samples, standard deviations should be indicated.

Appart from this, only minior corrections are needed:

  • Lines 24 and 32: Change "DDS 2205" to "DSS 2205"
  • Lines 135/136: The authors make refference to a one different welded sample but don't specify which one until line 103?, that sample should be identificated at line 96.
  • Figure 3 caption is not clear as figures 3b, 3d and 3f are identified as "HAZ", but they are also divided into 3 subimages: 1, 2 and 3, identified as "BM", "HAZ" and "WS".
  • Line 201: Please, describe the meaning of Ep where it appears for the first time.
  • There are some minor english mistakes along the paper.

Reviewer 4 Report

Dear Authors,

The reviewed submission titled: “Impacts of shielding gases on microstructures and properties of DSS2205 joints under solid wire metal insert gas welding“ is an experimental work on the study of mechanical corrosion properties of welded joints made of duplex steel, the most commonly grade used in the industry, made by the GMAW process. The work has a classic IMRaD layout, which perfectly corresponds to the content presented in it. The subject of the manuscript is in line with the current research trends in the weldability of two-phase steels and is in line with the scope of the Materials journal. In my opinion, the manuscript can be published, but it requires a few important additions and explanations, which I provide below:

General remark:

According to the current welding nomenclature, the MIG process (131) is not used for SS welding, as it is usually used as shielding gases - mixtures of inert and active gases (MAG process, 135 or FCAW, 136). Therefore, I propose to change (simplify and detail) the title of the work, for example: "Influence of shielding gas on microstructure and properties of GMAW DSS2205 welded joints." I suggest using the more general abbreviation GMAW (including MIG and MAG), because the scope of the research covers welding with both 100% Ar and chemically active mixtures (what is confirmed by the test results).

Authors names and surnames do not match those of the published work: Zha, X.-q.; Xiong, Y.; Zhou, T.; Ren, Y.-f.; Hei, P.-h.; Zhai, Z.-l.; Kömi, J.; Huttula, M.; Cao, W. Impacts of Stress Relief Treatments on Microstructure, Mechanical and Corrosion Properties of Metal Active-Gas Welding Joint of 2205 Duplex Stainless Steel. Materials 2020, 13, 4272. https://doi.org/10.3390/ma13194272

Please note that the term GMAW is used in this article.

Abstract should be redrafted according to the standard form: introductory sentence, purpose and scope of work, research performed, quantitative results and the most important conclusions.

Introduction:

Please use a citation method consistent with the journal's guidelines.

My general impression after reading this chapter is that it is too short to introduce the reader to the research part well and justify taking up the topic. In the literature, you can find many current articles describing the testing of weldability, mechanical and corrosion properties of duplex steels. In this regard, please consider relying on information from the works: https://doi.org/10.1002/maco.201709418, https://doi.org/10.1016/j.conbuildmat.2019.117697

In my opinion, this section lacks a deeper analysis of the state of the art. Research on the effect of gases on the properties of duplex steel joints is carried out by Dr. B. Varbai: https://scholar.google.com/citations?user=32Ysmb0AAAAJ&hl=pl&oi=ao

Lines 33-37: Please use the correct welding process names: 111 Manual Metal Arc Welding or Shielded Metal Arc Welding; 131 MIG Metal Inert Gas; 135 MAG-Metal Active Gas; 136 FCAW -Flux Cored Arc Welding; FCAW (MIG) is not used: flux cored wires are not currently produced for arc welding in inert gas (old process: 137).

Please clearly state the purpose of the work. Lines: 59-62: this is the work scope that should be described in chapter 2.

Chapter 2:

What were the dimensions (only thickness is given) of the joints?

Please provide the indication of electrode wire. Does the manufacturer recommend using a specific shielding gas?

Please provide the designations of the shielding gases. Were these commercially available gases?

Was it welded on both sides (double sided welding) in the PA position?

Add "arc" before "voltage".

Please provide the names of all devices used for testing (welding machine as well), names of manufacturers and their addresses (according to the Materials guidelines).

Line 125: change "minutes" to "min"

Add spaces before the units, correct the unit: mm2 (in line 138 should be mm).

Use the notation: "Figure" instead of "Fig."

Replace: “macrohardness" with"hardness".

Please add that a symmetrical distribution of the joint properties (with respect to the weld axis) was assumed. Figure 2 - please enlarge the font.

Chapter 3:

Correct the spelling of the name: Widmanstätten (twice).

What do the error bars show in the plots? Standard deviation?

Table 3: add spaces before the parentheses.

Chapter 3.2 should be titled: "Mechanical properties".

References must be formatted in accordance with the journal's guidelines. I propose to add current articles from the MDPI publishing house.

[9] - it is difficult to compare the effect of the type of shielding gas on the diffusible hydrogen content in welded joint on the basis of this publication, which concerns unalloyed steel. The metallurgy of welding high-alloy steels is completely different. Similar doubts arise from the analysis of the content of other cited works, eg: [13] and [19].

Round 2

Reviewer 2 Report

Dear Authors,

Thank you very much for the corrections made. 

Your paper in present form is suitable for publishing in my opinion.

Regards

Author Response

Thank you for your reply, I'm very happy that you could review my manuscript, and looking forward to a chance to contact you again!

Reviewer 4 Report

Dear Authors,

Thank you very much for all the explanations and professional additions to the content of the manuscript. I believe that the paper should be published, but I still have comments:

My comment from the first review: "Authors names and surnames do not match those of the published work" concerned the way of writing. Currently, compared to the previous article, the names and surnames are in the reverse order. It seems to me that Chinese surnames are usually monosyllabic. I am sorry for being unclear and I do not have sufficient knowledge in this regard, but a uniform record of authors' names is important for identifying their works in bibliographic databases. For example, WoS creates separate records in such a situation, which makes the query difficult.

The second issue: if some data concerning the welding procedure is confidential, please provide such information in the text.

Another thing: indeed, in some countries there are restrictions on access to internet resources, which is especially outrageous in the case of scientific information. The works of Dr. Varbai, which are related to the subject of the reviewed manuscript, for example:

https://www.sciencedirect.com/science/article/abs/pii/S0308016119300080

Finally, please format the bibliographic description of the references carefully.

Good luck!
